# Simulation and Nonlinearity Optimization of a High-Pressure Sensor

**DOI:** 10.3390/s20164419

**Published:** 2020-08-07

**Authors:** Ting Li, Haiping Shang, Weibing Wang

**Affiliations:** 1Institute of Microelectronics of The Chinese Academy of Sciences, Beitucheng West Road, Beijing 100029, China; liting@ime.ac.cn (T.L.); wangweibing@ime.ac.cn (W.W.); 2Department of Microelectronics, University of Chinese Academy of Sciences, 80 Zhongguancun East Road, Beijing 100049, China

**Keywords:** piezoresistive pressure sensor, nonlinearity, sensitivity, stress, simulation

## Abstract

A pressure sensor in the range of 0–120 MPa with a square diaphragm was designed and fabricated, which was isolated by the oil-filled package. The nonlinearity of the device without circuit compensation is better than 0.4%, and the accuracy is 0.43%. This sensor model was simulated by ANSYS software. Based on this model, we simulated the output voltage and nonlinearity when piezoresistors locations change. The simulation results showed that as the stress of the longitudinal resistor (R_L_) was increased compared to the transverse resistor (R_T_), the nonlinear error of the pressure sensor would first decrease to about 0 and then increase. The theoretical calculation and mathematical fitting were given to this phenomenon. Based on this discovery, a method for optimizing the nonlinearity of high-pressure sensors while ensuring the maximum sensitivity was proposed. In the simulation, the output of the optimized model had a significant improvement over the original model, and the nonlinear error significantly decreased from 0.106% to 0.0000713%.

## 1. Introduction

In recent years, pressure sensors have caused a lot of interest due to a wide spectrum of needs in various fields [1,2,3,4,5,6]. Compared with other common pressure sensors, such as piezoelectric pressure sensors, soft polymer pressure sensors, fiber Bragg grating pressure sensors, etc., silicon piezoresistive pressure sensors have the advantages such as small size, high integration, excellent performance, low cost, and high maturity. The development of Micro-Electro-Mechanical System (MEMS) and the expansion of its application, for example deep-sea survey and downhole exploration, have led to a rapid development of high-pressure sensors in recent years. In 2011, Zhao et al. designed and fabricated a high-pressure sensor in the range of 25 MPa with a circular diaphragm. They analyzed the effect of diaphragm size, shape, and piezoresistors positions on sensor performance by simulation [7]. The linearity is 0.08%, and the accuracy is 0.11%. In 2014, Niu et al. designed and manufactured an SOI (Silicon on Insulator) high-pressure sensor with a rectangular thick membrane structure [8], which can measure pressure up to 150 MPa, with a linearity of 0.3% and an accuracy of 0.48%.

Nonlinearity is one of the most important parameters of pressure sensors, which seriously affects the accuracy of pressure sensors, especially for wide-range pressure sensors. A lot of work has been done on nonlinearity optimization. In 1982, Yamada et al. showed through experiments and numerical analysis that there were optimum positions of the diffused layers for pressure sensors, which can effectively reduce the nonlinearity [9]. In 1995, Suzuki et al. studied the nonlinearity of square-diaphragm CMOS (Complementary Metal Oxide Semiconductor) integrated pressure sensors [10]. For any a/h value, a third-order approximation can be used to determine the optimum layout for piezoresistors to minimize their nonlinearity. This method uses numerical analysis under the large deflection theory and needs to measure the value of piezoresistive coefficients, and the calculation is very complicated. In 1989, Yasukawa et al. improved the linearity by changing the structure of the pressure-sensitive diaphragm to E-type or EI-type [11], and they also simulated these structures. In 2015, Nambisan et al. compared the conventional and bossed diaphragm pressure sensors by simulation [12]. Huang et al. designed a fan-shaped structure membrane with a central boss and optimized piezoresistors locations and nonlinearity with the help of simulations [13]. However, most of these researchers only conducted structural simulations without electrical simulations. They arranged the piezoresistors locations through the simulation results of stress to improve sensitivity and calculate the output voltage. In addition, the bossed diaphragm mainly reduces the geometric nonlinearity caused by the large deflection of the diaphragm. High-pressure sensors usually satisfy the small deflection theory due to a small ratio of side length to thickness of the diaphragm, so the nonlinearity mainly comes from the piezoresistive effect nonlinearity rather than geometric nonlinearity. Moreover, the fabrication process of the bossed diaphragm structure is more complicated. In 2014, Nisanth et al. compared the nonlinearity of pressure-sensitive diaphragms with different shapes by simulation [14]. They also only conducted structural simulations and only considered geometric nonlinearity, and changing the shape of the diaphragm only has a little improvement in linearity. In 2015, Zhang et al. optimized the linearity with a shallow boss structure and U-shaped piezoresistors [15]. They optimized the piezoresistors locations with the help of structural simulations, and the U-shaped resistors were not able to sufficiently optimize nonlinearity. Combining the theoretical analysis, structural simulations, and electrical simulations, we proposed a method under the small deflection theory for optimizing the nonlinearity of pressure sensors. This method is suitable for high-pressure sensors, and it is easy to calculate, implement, and fabricate. It can greatly optimize nonlinearity while having a good sensitivity.

## 2. Design and Fabrication of the Pressure Sensor

### 2.1. Design

A silicon-glass bonded absolute pressure sensor in the range of 0~120 MPa has been designed. It adopted a C-shaped silicon cup structure and square pressure-sensitive diaphragm. In order to reduce the geometric nonlinearity, the diaphragm meets the requirement of the small deflection theory, in which the maximum deflection is less than 1/5 of the diaphragm thickness [16,17],
(1)$$\omega_{max}<\frac{1}{5}h.$$

The diaphragm dimension is 272 μm × 272 μm × 160 μm. The substrate is N-type (100) silicon. 1.5 kΩ P-type piezoresistors are placed along <110> direction and form a Wheatstone bridge. Some piezoresistors cross the side of the pressure-sensitive diaphragm. The mask layout is shown in Figure 1. The output under constant voltage source excitation is [17]
(2)$$V_{out}=\frac{\Delta R}{R}V_{in}=\left(\pi_{l}\sigma_{l}+\pi_{t}\sigma_{t}\right)V_{in}\approx \frac{1}{2}\pi_{44}\left(\sigma_{l}-\sigma_{t}\right)V_{in}.$$

### 2.2. Fabrication

Figure 2 is the flow of the pressure sensor fabrication. First, a backside protective layer (SiO_2_ made by dry oxygen oxidation and SiN made by low-stress deposition) is deposited to prevent damage to the backside of the wafer during the subsequent process. Then, we implant boron ions to make piezoresistors. Next, we implant phosphorus ions to make the N+ isolation region, and then an ion implantation process is used to heavily dope boron ions to make a P+ low-resistance region adjoining the piezoresistors. Next, the wafer is annealed at 1100 °C for 90–150 min to achieve uniform doping in the top silicon. To connect the resistors to a Wheatstone bridge with aluminum, metal wiring is carried out by metal deposition and photolithography. A passivation layer (SiN) is deposited using PECVD (Plasma-Enhanced Chemical Vapor Deposition), and pads are etched using photolithography. Wet etching is used to etch the back cavity while leaving a diaphragm of 160 μm thickness. Finally, after removing the back protective layer, an anodic bonding process is used to bond 500 μm BF33 glass and MEMS silicon wafer to complete the fabrication of the pressure sensor chip.

Figure 3a,b are the top view and the cut view of the pressure sensor, respectively. The oil-filled package is used to complete the production of the pressure sensor. The chip is packaged in an independent oil-filled core. When measuring fluid pressure, the pressure directly acts on the corrugated diaphragm and is transmitted to the pressure sensor chip through the silicone oil. It can avoid direct contact between the measured liquid and the chip. It can also provide anti-overload protection to avoid problems such as liquid leakage caused by damage to the pressure-sensitive diaphragm. The oil-filled isolated package can improve the stability of the sensor and protect the MEMS chip. Figure 3c shows the packaged pressure sensor.

### 2.3. Static Performance Test

Due to the limitation of the test equipment and method, the test was conducted only within the range of 0–100 MPa, and the test results at room temperature are shown in Figure 4. Under the excitation of 1 mA constant current source, the zero-pressure output is 8.225 mV, and the output voltage is 29.4 mV when the pressure is 100 MPa. The sensitivity is 0.141 mV/(V MPa). The pressure sensor designed and fabricated by Jiang et al. has a range of 0–100 MPa, an output of 109 mV under the excitation of a 5 V constant voltage source, and a sensitivity of 0.22 mV/(V·MPa) [18]. Both sensors have a similar range and sensitivity. *NL* (*P*, *P_m_*) in Figure 4b represents the degree of nonlinearity,
(3)$$NL\left(P,P_{m}\right)=\frac{V\left(P\right)-V_{fit-line}\left(P\right)}{y_{FS}}.$$

The number with the maximum absolute value in *NL* (*P*, *P_m_*) represents the largest nonlinear degree, and this maximum absolute value is the nonlinear error. The nonlinear error is 0.788% when the end-point line is used as the fit-line. The linearity is 0.398% (the translated end-point line is used as the fit-line), repeatability is 0.144%, hysteresis is 0.096%, and accuracy is 0.434%.

## 3. Simulation and Comparison

*ANSYS* is used to set up the finite element analysis simulation. We build the same model as the experiment, and set the equivalent input voltage and the same applying pressure for simulation. In the simulation, the piezoresistive coefficients are π_11_ = 6.5, π_12_ = −1.1, π_44_ = 138.1 (10^−7^cm^2^/N), and the corresponding resistivity is 0.078 Ω·cm. The simulation model and resistors arrangement are shown in Figure 5a. The comparison between the simulation results and the experiment is shown in Figure 5b,c. The “Theoretical calculation” curve in the figure is calculated according to Formula (1) and the stress simulation result (“σ_l_-σ_t_” curve in Figure 5b). We draw the “Experiment (origin)” curve by translating the “Experiment” output curve to the origin.

The simulation output is almost completely consistent with the theoretical calculation. The output of the experiment is smaller than the simulation. The first reason is that a higher doping concentration of the experimental specimen leads to smaller piezoresistive coefficients than the simulation. The second reason is that the experiment is not in the ideal situation as the simulation; there is a certain difference in the stress. The third reason is that the low-resistance region and metal wiring omitted in the simulation and the parasitic resistance and capacitance in the actual circuit will also influence the output of the experiment. In addition, there is a zero drift in the experiment, which may be related to the asymmetry etching and nonuniform doping of piezoresistors in the fabrication. The nonlinear error of the experimental specimen is 0.788%, and the nonlinear error in the simulation is 0.0845% (these are the calculation results when the end-point line is used as the fit-line). The nonlinear error of the actual specimen is about 9 times the simulation result, which is mainly because the simulation conditions are very ideal, while the experiment is affected by many other factors, such as nonuniform doping, asymmetry of the pattern etching, etc., especially the nonlinearity caused by the package, which has a great influence. In general, the simulation and experiment have the same trend. So, the simulation results have a certain reference value. When pressurized to 120 MPa, the nonlinear error in the simulation is 0.106% (Figure 10c), and the nonlinear error of the specimen is estimated to be about 0.954%. If the end-point translation line is used as the fit-line, the nonlinear error is about 0.48%.

## 4. Nonlinearity Optimization

### 4.1. Theoretical Analysis

Only the linear part of the piezoresistive effect is considered in Formula (1). In fact, the piezoresistive effect is only approximately linear. Research on the piezoresistive effect of a single resistor shows that third-order stress terms are sufficient to give a good approximation [19,20],
(4)$$\frac{\Delta R}{R_{0}}=\pi_{1}T+\pi_{2}T^{2}+\pi_{3}T^{3}$$
where *π*_1_, *π*_2_, and *π*_3_ are the first-order, second-order, and third-order piezoresistive coefficients, respectively, and *T* is stress. Our research group has studied the method of an asymmetric Wheatstone bridge to optimize nonlinearity [21], and we found that when the resistance of *R_L_* (the resistor perpendicular to the near side) is greater than *R_T_* (the resistor parallel to the near side), the nonlinear error will first decrease to around 0 and then increase, so when *R_L_* > *R_T_* and the difference is within a certain range, the linearity of the device can be improved, but additional zero output will be generated, which is not expected. On the other hand, when the diaphragm is pressed, *R_L_* increases and *R_T_* decreases, and the rate of change is related to the change of the stress at the resistor’s positions. The sensor diaphragm satisfies the small deflection theory, so the stress and pressure show a good linear relationship [16,17], assumed that,
(5)T=KP.

For different positions on the diaphragm, the stress is different under the same pressure; that is, *K* is different. Therefore, *R_L_* and *R_T_* can be placed in asymmetric stress positions to achieve the purpose of *R_L_* > *R_T_*, thereby improving the nonlinearity and eliminating additional zero output.

### 4.2. Simulation

Firstly, the stress at different positions on the pressure-sensitive diaphragm is simulated. Then, according to the simulation result of the stress distribution, *R_L_* and *R_T_* are placed at positions with the same stress for simulating. Next, we change the location of *R_L_* and study the change of the nonlinearity when *R_L_* and *R_T_* are at different stress positions. Assume that at the initial piezoresistors locations, *T = KP*, at the changed *R_L_* location, *T* = (*K* + Δ*K*) *P*. In order to determine the locations of resistors more accurately, the U-shaped resistors are replaced by strip-shaped resistors in this simulation, as shown in Figure 6.

### 4.3. Results

Figure 7a is the stress simulation result of different positions on the diaphragm. Figure 7b–d show the simulation results when *R_L_*’s stress position changes. The largest nonlinear degree usually occurs at *P = P_m_*/2, but when it is extremely small, the corresponding *P* will change. When Δ*K* increases, the largest nonlinear degree increases from negative to positive, so the nonlinear error reduces to about 0 and then increases.

### 4.4. Results Discussion

According to Expressions (4) and (5), *R_L_* and *R_T_* are respectively expressed as,
(6)$$R_{L}=R_{0}+\left\{\pi_{L1}\left(K+\Delta K\right)P+\pi_{L2}{\left[\left(K+\Delta K\right)P\right]}^{2}+\pi_{L3}{\left[\left(K+\Delta K\right)P\right]}^{3}\right\}R_{0}$$
(7)$$R_{T}=R_{0}+\left[\pi_{T1}KP+\pi_{T2}{\left(KP\right)}^{2}+\pi_{T3}{\left(KP\right)}^{3}\right]R_{0}.$$

The subscript *L* means longitudinal and *T* means transverse, and 1, 2, 3 mean first-order, second-order, and third-order piezoresistive coefficients, respectively. According to the calculation, the Wheatstone bridge output is,
(8)$$V_{out}=\frac{R_{L}-R_{T}}{R_{L}+R_{T}}V_{in}.$$

So, the zero-pressure output is
(9)$$V_{out0}=0.$$

In following expressions, assume that
(10)$$A\left(x\right)=\left(\pi_{L1}-\pi_{T1}\right)Kx+\left(\pi_{L2}-\pi_{T2}\right)K^{2}x^{2}+\left(\pi_{L3}-\pi_{T3}\right)K^{3}x^{3}$$
(11)$$B\left(x\right)=\left(\pi_{L1}+\pi_{T1}\right)Kx+\left(\pi_{L2}+\pi_{T2}\right)K^{2}x^{2}+\left(\pi_{L3}+\pi_{T3}\right)K^{3}x^{3}.$$

So, the output at pressure *P* is
(12)$$V_{out}\left(P\right)=\frac{\left(\pi_{L1}P+\pi_{L2}KP^{2}+3K^{2}P^{3}\right)\Delta K+\left(\pi_{L2}P^{2}+3KP^{3}\right)\Delta K^{2}+P^{3}\Delta K^{3}+A\left(P\right)+K^{3}P^{3}}{\left(\pi_{L1}P+\pi_{L2}KP^{2}+3K^{2}P^{3}\right)\Delta K+\left(\pi_{L2}P^{2}+3KP^{3}\right)\Delta K^{2}+P^{3}\Delta K^{3}+2+B\left(P\right)+K^{3}P^{3}}V_{in}.$$

The full-scale output is
(13)$$Y_{FS}=\frac{\left(\pi_{L1}P_{m}+\pi_{L2}KP_{m}{}^{2}+3K^{2}P_{m}{}^{3}\right)\Delta K+\left(\pi_{L2}P_{m}{}^{2}+3KP_{m}{}^{3}\right)\Delta K^{2}+P_{m}{}^{3}\Delta K^{3}+A\left(P_{m}\right)+K^{3}P_{m}{}^{3}}{\left(\pi_{L1}P_{m}+\pi_{L2}KP_{m}{}^{2}+3K^{2}P_{m}{}^{3}\right)\Delta K+\left(\pi_{L2}P_{m}{}^{2}+3KP_{m}{}^{3}\right)\Delta K^{2}+P_{m}{}^{3}\Delta K^{3}+2+B\left(P_{m}\right)+K^{3}P_{m}{}^{3}}V_{in}.$$

Considering that the nonlinearity is mainly compensated by the difference in resistance caused by different stresses, and the difference in resistance is mainly caused by the first-order effect of the change in the stress, so the effect of Δ*K* on the change of the resistance higher-order terms is omitted. The degree of nonlinearity at pressure *P* with reference to the maximum pressure *P_m_* can be simplified as
(14)$$NL\left(P,P_{m}\right)=\frac{V\left(P\right)-V_{fit-line}\left(P\right)}{Y_{FS}}=\frac{\pi_{L1}^{2}PP_{m}\Delta K^{2}+\left\{\pi_{L1}P_{m}A\left(P\right)+\pi_{L1}P\left[2+B\left(P_{m}\right)\right]\right\}\Delta K+A\left(P\right)\left[2+B\left(P_{m}\right)\right]}{\pi_{L1}^{2}PP_{m}\Delta K^{2}+\left\{\pi_{L1}PA\left(P_{m}\right)+\pi_{L1}P_{m}\left[2+B\left(P\right)\right]\right\}\Delta K+A\left(P_{m}\right)\left[2+B\left(P\right)\right]}-\frac{P}{P_{m}}.$$

When *P* and *P_m_* are determined and it is assumed that *P = P_m_*/2, the full-scale output and *NL (P,P_m_)* can be expressed as the following form expression,
(15)$$Y_{FS}=\frac{\Delta K+a\Delta K^{2}+b\Delta K^{3}+c_{1}}{\Delta K+a\Delta K^{2}+b\Delta K^{3}+c_{2}}V_{in}$$
(16)$$NL\left(\%\right)=NL\left(\frac{P_{m}}{2},P_{m}\right)=\frac{\Delta K^{2}+d_{1}\Delta K+e_{1}}{\Delta K^{2}+d_{2}\Delta K+e_{2}}-\frac{1}{2}$$
where *a*, *b*, *c*_1_, *c*_2_, *d*_1_, *d*_2_, *e*_1_, and *e*_2_ are parameters related to *P_m_*. Use *Origin 9.1* to fit the simulation results with Expressions (15) and (16), and the fitting results are shown in Figure 8.

The *Adj. R-Square* in the two fitting results are 0.99866 and 0.99902 respectively, indicating that the simulation results are almost consistent with the theoretical calculation. The full-scale output increases with increasing Δ*K* because *R_L_* has moved to an area where the stress changes greater. So, the pressure sensor sensitivity increases. The nonlinearity of the device is compensated at the circuit level due to the difference in resistance caused by the different *R_L_* and *R_T_* stress positions, so the nonlinearity is optimized. In addition, the nonlinear error is the maximum absolute value in the calculation results of all test points, so it can be close to but not equal to 0. Only when the nonlinear error is extremely small, the corresponding *P* is not *P_m_*/2, so Equations (15) and (16) can still be used to estimate the difference in the stress at the piezoresistors locations to get optimal linearity.

## 5. A Method for Optimizing Nonlinearity of Piezoresistive Pressure Sensors

According to the above conclusion, a method for optimizing the nonlinearity of the pressure sensor is proposed. Firstly, place *R_L_* in the area with the maximum stress, and then adjust the location of *R_T_* appropriately so that it is at a slightly lower stress position than the location of *R_L_*, as shown in Figure 9.

Through this method, the model of the pressure sensor fabricated above was optimized and simulated. The optimized model is shown in Figure 10a, in which the piezoresistors locations change. The simulation results are shown in Figure 10b,c.

In the simulation results, the nonlinear error is reduced from 0.106% to 0.0000713%, and the linearity is improved significantly by 4 orders of magnitude. According to the comparison of the experimental specimen and simulation results, it is estimated that the linearity can reach 0.00064%. The linearity of pressure sensors without circuit compensation hardly reach this order of magnitude. The pressure sensor in the range of 0–150 MPa has a linearity of 0.3% [8], and the pressure sensor in the range of 100 MPa with compensation resistors has a linearity of 0.07% [18]. Furthermore, the maximum output voltage of the original model is 42.2 mV, and its sensitivity is 0.234 mV/(V·MPa). While the maximum output voltage of the optimized model is 71.4 mV, and its sensitivity is 0.397 mV/(V·MPa). It is estimated that the sensitivity in the experiment can reach 0.24 mV/(V·MPa). The sensor has a higher range than 100 MPa [18], while has a slightly higher sensitivity. The sensitivity has also been significantly improved.

## 6. Conclusions

A wide measurement range pressure sensor in the range of 0–120 MPa was designed and fabricated with a linearity of 0.4% and an accuracy of 0.43%. We found through simulation that the nonlinearity of the pressure sensor will change if *R_L_* and *R_T_* are at asymmetric positions on the pressure-sensitive diaphragm. Assuming at the location of *R_T_*, *T = KP*, and at the location of *R_L_*, *T* = (*K* + Δ*K*) *P*, as Δ*K* increases, the nonlinear error will decrease to around 0 and then increase. Based on this discovery, a method for optimizing the nonlinearity of the pressure sensor is proposed while ensuring maximum sensitivity. The *R_L_* is placed at the position with the maximum stress; then, the *R_T_* is adjusted to a position where the stress is slightly smaller, which can effectively reduce the nonlinear error with a good sensitivity. The pressure sensor is optimized by this method, and the simulation results show that the linearity of the model has been significantly improved after optimization.

## Figures and Tables

**Figure 1 sensors-20-04419-f001:**
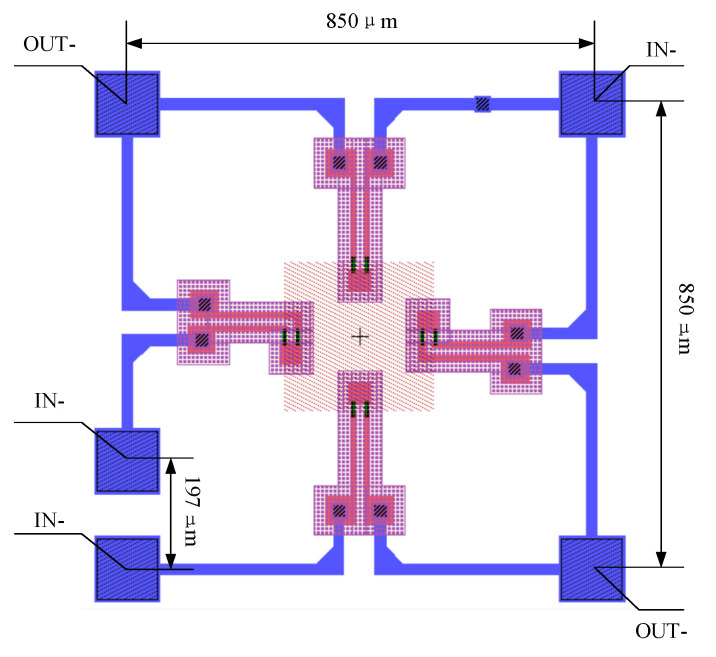
Mask layout of the pressure sensor.

**Figure 2 sensors-20-04419-f002:**
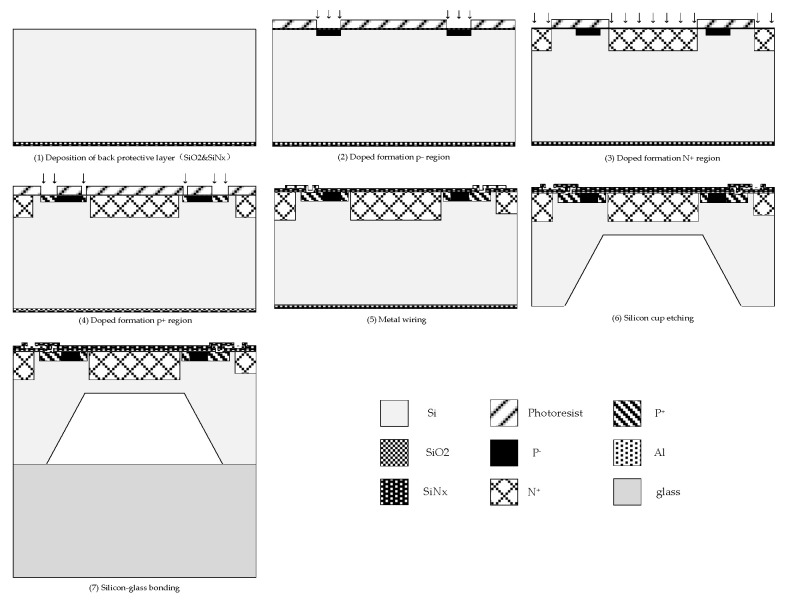
Main fabrication process of the pressure sensor.

**Figure 3 sensors-20-04419-f003:**
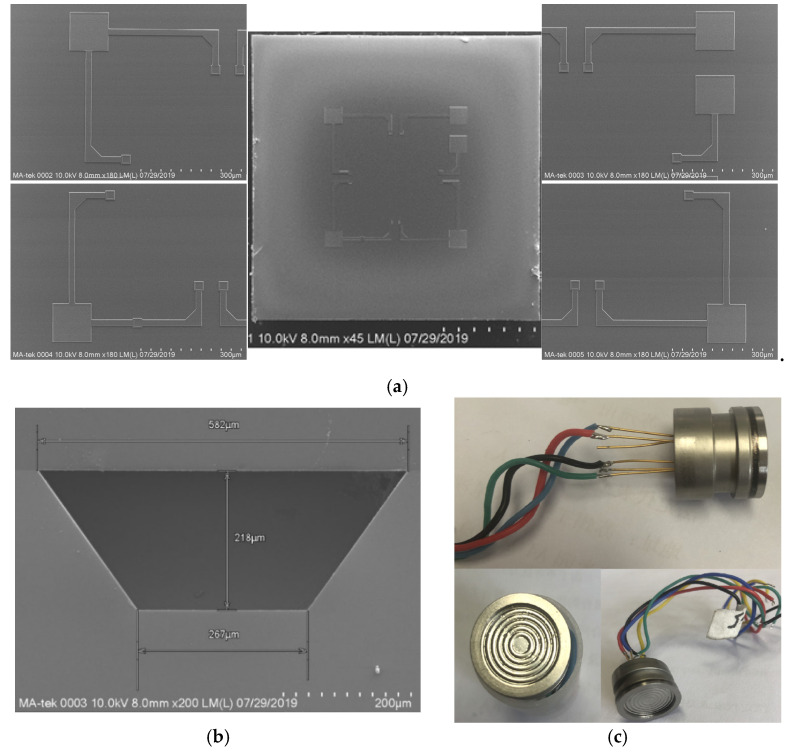
(**a**) Top view of the chip; (**b**) Cut view of the chip; (**c**) Packaged pressure sensor.

**Figure 4 sensors-20-04419-f004:**
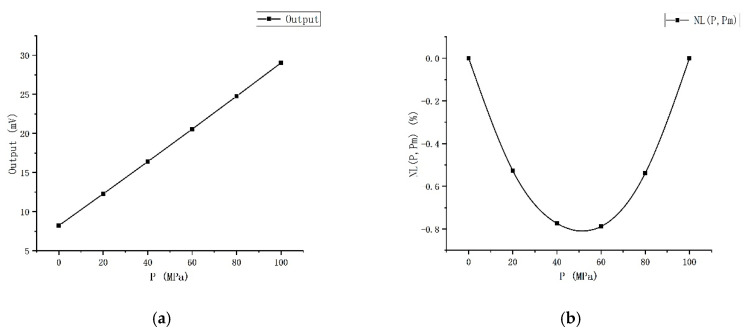
Static performance test results: (**a**) Output voltage; (**b**) Nonlinearity.

**Figure 5 sensors-20-04419-f005:**
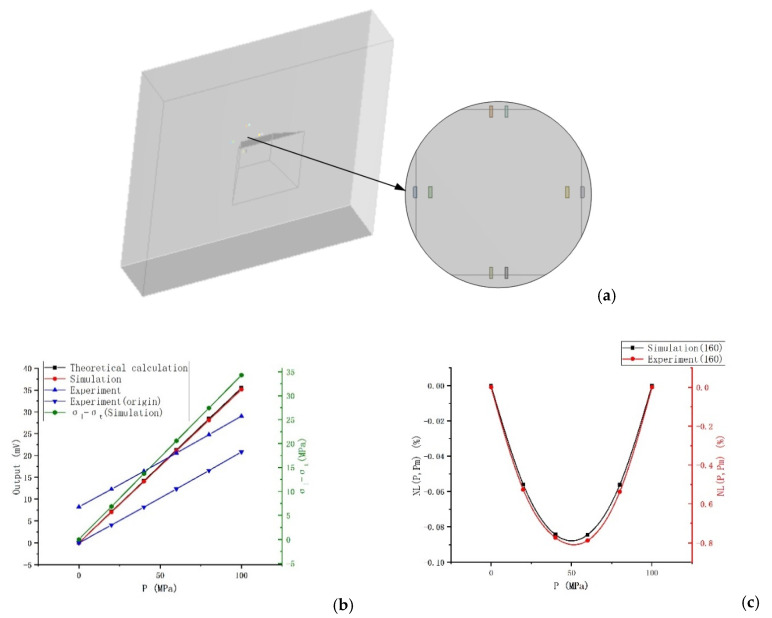
Simulation and experiment: (**a**) The model and resistors arrangement; (**b**) Output voltage; (**c**) Nonlinearity.

**Figure 6 sensors-20-04419-f006:**
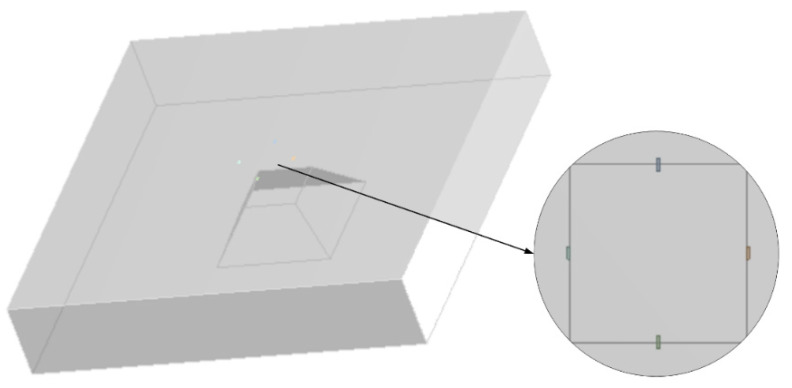
Simulation model.

**Figure 7 sensors-20-04419-f007:**
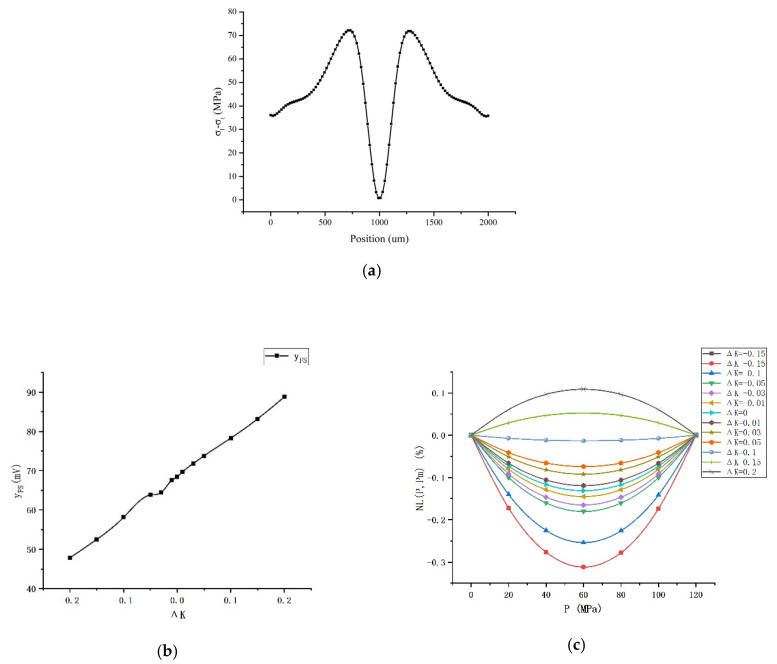
Simulation results: (**a**) Stress–position relationship; (**b**) The changes in full-scale output; (**c**) The changes in *NL* (*P,P_m_*); (**d**) The changes in the largest nonlinear degree; (**e**) The changes in nonlinear error.

**Figure 8 sensors-20-04419-f008:**
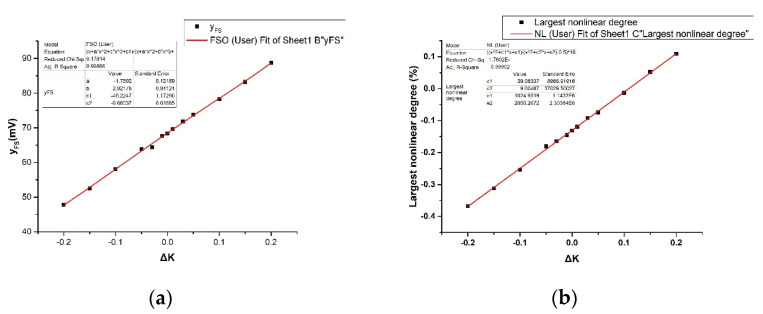
Simulation results fitting: (**a**) Full-scale output changes fitting; (**b**) Largest nonlinear degree changes fitting.

**Figure 9 sensors-20-04419-f009:**
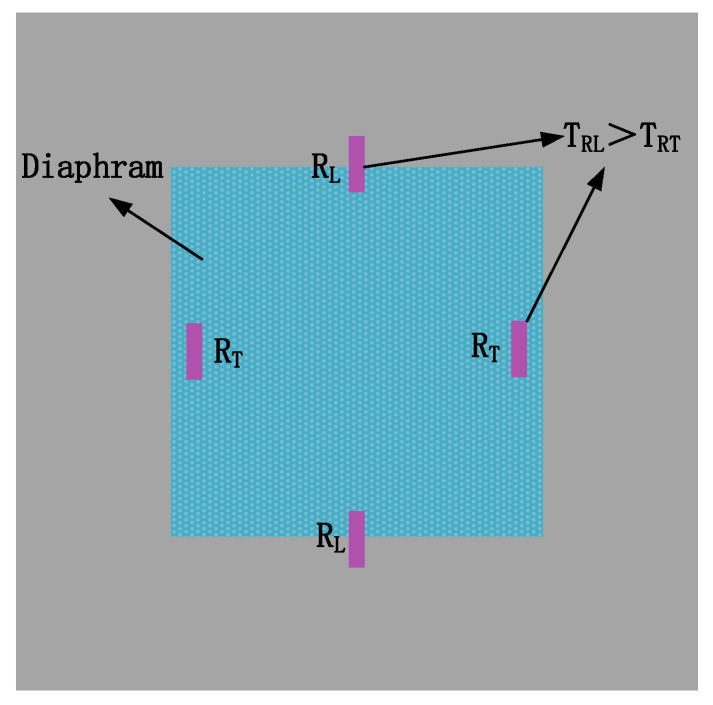
A method for optimizing nonlinearity of the pressure sensor.

**Figure 10 sensors-20-04419-f010:**
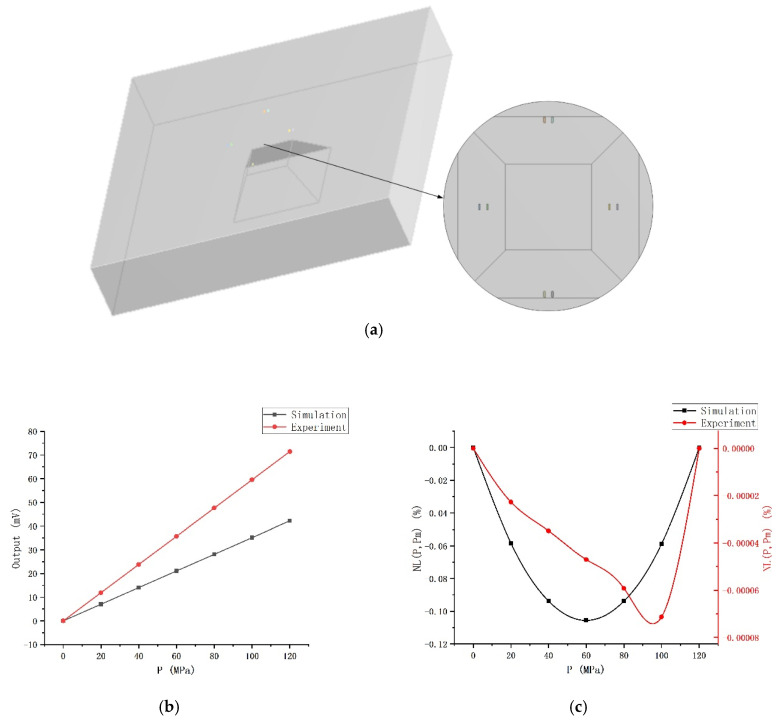
Simulation results of the optimized model: (**a**) The optimized model; (**b**) Output voltage; (**c**) Nonlinearity.

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
