# Peer review of "Simulation and Nonlinearity Optimization of a High-Pressure Sensor"

_sensors, 2020, doi:10.3390/s20164419_

Round 1
Reviewer 1 Report
-
The paper must be proofread for English.
-
No discrimination and scientific advances compared to existing research
- No improvement in nonlinearity, repeatability and hysteresis compared to Ref [2]
- Previous studies show the effectiveness of pressure measurement at high temperatures (more about 200 degrees), but temperature-related experimental conditions are not described in the paper.
Author Response
Dear Reviewer,
Thank you for your comments concerning our manuscript. Those comments are all valuable and very helpful for revising and improving our paper, as well as the important guiding significance to our researches. We have studied comments carefully and have made correction which we hope meet with approval. Revised portion are marked in red in the paper.
The responds to the comments are as follows:
- Response to comment: The paper must be proofread for English.
Response: We are very sorry for our incorrect writing. We have proofread the article for English and carefully revised the spelling and grammar errors. - Response to comment: No discrimination and scientific advances compared to existing research.
Response: It is really true as Reviewer suggested that there are no discrimination and scientific advances compared to existing research. The main reasons are as follows: This work aims to manufacture a 120MPa high pressure sensor, and proposes a new nonlinear optimization method through simulation and test comparison analysis, which is used to optimize the nonlinearity of piezoresistive pressure sensors. The test results are mainly used to prove the reliability of the simulation and to propose optimization schemes on this basis. So the pressure sensor in this paper have no significant advantages in performance. - Response to comment: No improvement in nonlinearity, repeatability and hysteresis compared to Ref [2].
Response: We are very sorry for our negligence of the error in the description of linearity for Ref [2] (now Ref [8]). We have corrected it on line 36. Compared with Ref [8], the nonlinearity, repeatability, and hysteresis of the sensor in this paper did not improve in value. The main reason may be that the pressure sensor in Ref [8] has some circuit compensation, which is conducive to improving the performance of the chip, but we did not. In addition, this work aims to manufacture a 120MPa high pressure sensor, and proposes a new nonlinear optimization method through simulation and test comparison analysis, which is used to optimize the nonlinearity of piezoresistive pressure sensors. Therefore, the linearity, repeatability and hysteresis of the pressure sensors currently fabricated are not optimal. - Response to comment: Previous studies show the effectiveness of pressure measurement at high temperatures (more about 200 degrees), but temperature-related experimental conditions are not described in the paper.
Response: It is the deep-sea measurement that the high-pressure sensor designed and fabricated in this paper is used for. the ambient temperature range of the deep-sea is between -5℃ and 35℃, so no high-temperature pressure test has been conducted. It has been added on line 103 of the article that the test temperature is room temperature.
Special thanks to you for your good comments!
Reviewer 2 Report
This work described the fabrication of a high-pressure sensor and the optimization of Nonlinearity. The manuscript is well presented. I suggest that the work is suitable for publication in Sensors after some minor revision. Other comments to this work as following;
(1) More Figures are listed in manuscript. Some of them should be combined. For example. figure 3 and figure 4, figure 6 and figure 7, Figure 9 and figure 10, figure 12 and figure 13 should be combined.
(2) English language should be smoothed in detail, some spelling errors must be corrected, for example, Optimizaton in title should be Optimization.
Author Response
Dear Reviewer,
Thank you for your comments concerning our manuscript. Those comments are all valuable and very helpful for revising and improving our paper, as well as the important guiding significance to our researches. We have studied comments carefully and have made correction which we hope meet with approval. Revised portion are marked in red in the paper.
The responds to the comments are as follows:
- Response to comment: More Figures are listed in manuscript. Some of them should be combined. For example. figure 3 and figure 4, figure 6 and figure 7, Figure 9 and figure 10, figure 12 and figure 13 should be combined.
Response: Considering the Reviewer’s suggestion, we have combined some of the pictures in the paper, for example the simulation model and results. Figures 3, 5, 7, and 10 in the revised paper are the combined results. - Response to comment: English language should be smoothed in detail, some spelling errors must be corrected, for example, Optimizaton in title should be Optimization.
Response: We are very sorry for our incorrect writing. We have proofread the article for English, carefully revised the spelling and grammar errors, and smoothed the English language in detail.
Special thanks to you for your good comments!
Reviewer 3 Report
This paper acts to suggest the development of a MEMS micro diaphragm pressure sensor. Both simulation and experiments have been conducted to study the performance of the proposed sensor. ALothough it is an interesting paper, details provided for fabrication packaging and testing are not sufficient. Authors need to expand more on their results. Below are the suggestions and comments that need to be applied.
1) Line 26, MEMS need to be defined. Please make sure all the acronyms have been defined in their full term when they are first used in the paper.
2) The introduction needs to be more complete. MEMS micro diaphragm piezoelectric pressure sensors have been widely used for various applications https://doi.org/10.1088/0960-1317/24/1/015017. Polymer-based micro diaphragm pressure/flow sensors have been preferred over silicon-based MEMS pressure sensors https://doi.org/10.1177/1045389X14521702. The authors need to comment on these sensors in the introduction.
3) Fabrication process in figure 2 needs to be explained with more details.
4) What type of bonding has been used for the silicon-glass bonding process? What were the challenges of this step?
5) The quality of figure 3a is very poor. Please change it with a sharper image.
6) How did you come up with the packaging scheme shown in figure 4?
7) The sensor output in Figure 5 is based on Voltage change. How did you convert the resistance change to Voltage output?
8) Results presented in Figures 5 and 7 requires error bars. How many times did you repeat the experiments?
9) The sensitivity and working range (span) of the proposed sensor need to be compared with similar MEMS sensors previously published in the literature.
10) There are some grammatical errors in the paper. Manuscript need to check to ensure it is error-free
Round 2
Reviewer 3 Report
Authors have applied all the comments. I suggest this paper can be published in its present form.
Author Response
Dear Reviewer,
Thank you very much for your positive and constructive comments on our manuscript. We have made revision which marked in red in the paper.
Once again, thank you for your warm work.